# Prevotella: A Key Player in Ruminal Metabolism

**DOI:** 10.3390/microorganisms11010001

**Published:** 2022-12-20

**Authors:** Claudia Lorena Betancur-Murillo, Sandra Bibiana Aguilar-Marín, Juan Jovel

**Affiliations:** 1Escuela de Ciencias Básicas, Tecnología e Ingeniería, Universidad Nacional Abierta y a Distancia, UNAD, Bogotá 111511, Colombia; 2Facultad de Ciencias Agropecuarias, Universidad de Caldas, Manizales 170004, Colombia; 3Faculty of Veterinary Medicine, University of Calgary, 3280 Hospital Dr NW, Calgary, AB T2N 4Z6, Canada

**Keywords:** *Prevotella*, carbohydrate metabolism, propionate, methane emissions, sustainable agriculture

## Abstract

Ruminants are foregut fermenters that have the remarkable ability of converting plant polymers that are indigestible to humans into assimilable comestibles like meat and milk, which are cornerstones of human nutrition. Ruminants establish a symbiotic relationship with their microbiome, and the latter is the workhorse of carbohydrate fermentation. On the other hand, during carbohydrate fermentation, synthesis of propionate sequesters H, thus reducing its availability for the ultimate production of methane (CH4) by methanogenic archaea. Biochemically, methane is the simplest alkane and represents a downturn in energetic efficiency in ruminants; environmentally, it constitutes a potent greenhouse gas that negatively affects climate change. *Prevotella* is a very versatile microbe capable of processing a wide range of proteins and polysaccharides, and one of its fermentation products is propionate, a trait that appears conspicuous in *P. ruminicola* strain 23. Since propionate, but not acetate or butyrate, constitutes an H sink, propionate-producing microbes have the potential to reduce methane production. Accordingly, numerous studies suggest that members of the genus *Prevotella* have the ability to divert the hydrogen flow in glycolysis away from methanogenesis and in favor of propionic acid production. Intended for a broad audience in microbiology, our review summarizes the biochemistry of carbohydrate fermentation and subsequently discusses the evidence supporting the essential role of *Prevotella* in lignocellulose processing and its association with reduced methane emissions. We hope this article will serve as an introduction to novice *Prevotella* researchers and as an update to others more conversant with the topic.

## 1. Introduction

The genus *Prevotella* was named after the French microbiologist André Romain Prévot [1]. It belongs to the Prevotellaceae family, which also includes the kindred genera *Paraprevotella*, *Alloprevotella* and *Hallella* [2]. The genus *Prevotella* comprises more than 50 anaerobic, non-spore-forming, Gram-negative species which are largely saccharolytic [3], and generate short-chain fatty acids (SCFAs) as fermentation products [4,5]. *Prevotella* is ubiquitous. For example, it is found in various body environments (oral cavity, urogenital and gastrointestinal tracts) [3,6,7,8,9,10] and across multiple species of animals, including humans, livestock, rodents and insects [1,2,3,4,11,12,13,14,15,16]. In the phylum Bacteroidetes, the phylogenetic and biological distinction between the genera *Bacteroides* and *Prevotella* is rather unclear [17,18], and this explains apparent controversies found in the early literature. Consequently, many former *Bacteroides* spp. were reclassified as *Prevotella*, *Porphyromonas*, *Parabacteroides* and *Alistipes* [19].

The rumen microbiome includes bacteria, ciliate protozoa, archaea, fungi and viruses, and their abundance follows the same order [20,21,22,23,24]. As in other organisms, the prevalent bacteria phyla in the rumen are Bacteroidetes and Firmicutes (synonyms Bacteroidota and Bacillota, respectively) which include a series of fiber-degrading bacteria [25,26,27]. *Prevotella* is often reported as a dominant genus of the rumen microbiome [28]. For example, in a comprehensive study, including 742 animals from 35 countries and 32 species of ruminants and camelids, Henderson and collaborators reported that the most abundantly identified taxa corresponded to members of the genera *Prevotella*, *Ruminococcus*, and *Butyrivibrio,* as well as unclassified members of the orders Clostridiales and Bacteroidales and of the families Ruminococcaceae and Lachnospiraceae [29]. Another large study including seven farms and 816 Holstein cows from the UK and Italy and 200 Nordic Red cows from Sweden and Finland also reported Bacteroidales, Spirochaetales, and WCHB1-41 as the most common bacterial taxa, and all operational taxonomic units (OTUs) found to be heritable in the Nordic red cohort were related to the Prevotellaceae family [26]. Moreover, a study including 695 samples from eight sites in the four compartments of the stomach of buffaloes also reported *Prevotella* among the most abundant taxa [30]. *Prevotella* was also shown as one of the most frequent taxa in a study that assembled 10,000 ruminal metagenomes [31]. A series of other studies with smaller cohorts also hinted towards the important role of Prevotellaceae as ruminal bacteria in cattle [18,32,33,34], buffalo [35,36,37,38,39], sheep [12,40,41,42,43], goat [11,44,45,46], domestic yak [47,48,49,50] and deer [51,52,53].

Over half a century ago, Robert Hungate and collaborators put forward the notion that the rumen microbiome is highly influenced by ingested feedstuff, a selection process driven by microbial biochemical efficiency, and the incorporation of new microbes into niches created by highly competitive microorganisms displacing less-competitive ones [20]. Thus, the abundance of *Prevotella* in metagenomic studies spanning several species of ruminants is perhaps a reflection of their biochemical efficiency and more generally of their adaptation to the rumen environment. As shown below, genetics of the host and its interaction with the microbiome are also important in determining the microbiome’s population structure and its metabolic efficiency [54,55].

*Prevotella* is central to carbohydrate and hydrogen metabolism and high abundance of *Prevotella* in ruminants is associated with a healthy microbiome [36,56,57,58,59,60,61,62,63]. As found in other Bacteroidetes and more generally in fiber-degrading bacteria, the *Prevotella* genome is endowed with polysaccharide utilization loci (PUL), which are gene clusters that encode proteins specialized in the processing of complex carbohydrates [3,64,65,66]. Ruminants ingest insufficient amounts of glucose in their diet, and to compensate for this limiting nutrient they engage in gluconeogenesis. *Prevotella* is able to break down a variety of polysaccharides and has the capacity to synthesize propionate, which in turn is the most important substrate for gluconeogenesis in the liver of ruminants [4,5,12,59,67]. Metabolic profiling of the *Prevotella* genome revealed enrichment of roles that include amino acid, carbohydrate, lipid, cofactors and vitamins, nucleotide and energy (ATP) metabolism, genetic information processing, membrane transport, replication and repair and translation [1,3,13,36,68,69,70]. This is likely a reflection of adaptations to an ecological niche where carbohydrates and free amino acids are limiting factors.

One of the end products of ruminal fermentation is methane (CH_4_) which is a contaminant greenhouse gas and represents a loss of energy for animals [71,72,73]. Thus, procuring a rumen microenvironment that minimizes methane production is energetically and environmentally favorable. Central to rumen fermentation is the intra- and extracellular flux of hydrogen, which can be diverted between the production of fatty acids or methane [74]. Hydrogen production is an intrinsic problem of ruminal fermentation. More than half of sequenced ruminal microbes encode hydrogenases in one or more of 26 hydrogenases subgroups. In a meta-study that assembled 10,000 bacterial metagenome-assembled genomes (MAGs) more than 6000 of those MAGs contained genes encoding [NiFe]-, [FeFe]-, and Fe-hydrogenases, with more than 3000 encoding enzymes for production of H_2_, mostly in Firmicutes, and 95 additional MAGs encoded hydrogenases for H_2_-uptake and methyl-CoM reductases for methanogenesis [31]. In metatranscriptomic experiments in the sheep rumen, half of electron bifurcating [FeFe]-hydrogenases expressed belonged to the class Clostridia. Interestingly, several H_2_-uptake pathways, including methanogenesis, fumarate and nitrite reduction, and acetogenesis were found to be differentially expressed in sheep with high and low methane emissions. This opens opportunities for experimental manipulation of the microbiome to favor expression of pathways that compete with methanogenesis [75,76]. Descriptive and comparative metagenomic studies in multiple species of ruminants have revealed that *Prevotella* is not only beneficial for efficient biosynthesis of nutrients by ruminants but also significantly ameliorates negative effects of ruminal metabolism on the environment. In this article, we summarize the contribution of *Prevotella* to ruminal metabolism and their association with lower emissions of methane derived from carbohydrate fermentation.

## 2. Ruminal Carbohydrate Fermentation

Plant tissues are mostly carbohydrates that constitute the main source of energy for ruminants and their commensal microorganisms [54,77,78,79]. Carbohydrates are structural components of the plant cell wall, but also exist as intracellular pools of carbon [80,81]. The digestive system of ruminants is complex (Figure 1A); in addition to the small and large intestines, the pancreas, and the gallbladder, there is a stomach with four compartments: the rumen, reticulum, omasum and abomasum [82]. The fermentation process starts with chewing of forage or concentrate to reduce the size of particles and to increase the surface area of the feed that will make contact with the aqueous environment of the rumen, where fermentative microbes intercept feedstuff [21,83]. The first two components of the ruminant stomach (rumen and reticulum) constitute the forestomach or fermentation vat [82]. Scratching of the rumen surface by fibers induce vigorous contractions of the rumen, which helps mixing the ingesta, but the reticulum also experiences contractions that further mix feedstuff and help regurgitation of feed boluses that need remastication for more efficient fermentation (Figure 1A) [21,60,61,84]. Indeed, it has been demonstrated that feedstuff surface area is a more important parameter of fermentation efficiency than the crystallinity of plant materials [21].

The anaerobic nature of the rumen provides a suitable niche for microorganisms that act as workhorses for carbohydrate fermentation [20,21,61,83,87]. Bacterial density in the rumen may be as high as 10^10^ cells per g or ruminal content [21]. Nutrients are extracted by microbes under thermodynamically favorable conditions in an aqueous solution rich in bicarbonate, contributed by the saliva of animals, which buffers acidity in the rumen [21]. It has been reported that saliva production in adult cows may exceed 200 L per day [88]. An anaerobic environment also ensures only partial oxidation of carbohydrates to CO_2_ and H_2_O and enables enzymatic extraction of energy by microorganisms in the form of intermediate energetic molecules called volatile fatty acids (VFAs), most commonly propanoic acid, acetic acid, and butyric acid [74,84]. Maintenance of constant and favorable temperature and pH for microbial activity in the rumen is essential to achieve optimal metabolic efficiency [89]. Appropriate temperature is maintained by the exergonic nature of fermentation itself (flux of electrons during fermentation) [75], while VFAs should be continuously mobilized through the ruminal epithelium to the portal vein, and finally to the liver, to be distributed but also to prevent excessive pH reduction in the ruminal chamber [21].

What follows is a simplified description of the biochemical reactions during carbohydrate fermentation (Figure 1B). For a more complete overview, the reader is encouraged to read several excellent articles published by experts in this field [21,60,74,77,83,84,90,91,92]. Lignocellulose is the most abundant carbohydrate-containing compound on Earth, as it constitutes the dry matter in vegetation. Lignocellulose is composed of two types of polymers, cellulose and hemicellulose that are the core feedstuffs of ruminants. Additionally, livestock fed with processed concentrates derived from cereals (wheat, corn, barley, etc.) ingests substantial amounts of starch. All three (cellulose, hemicellulose and starch) contain hexose, a sugar molecule with six carbon atoms (C_6_H_12_O_6_), which through glycolysis (EMP pathway) is converted into pyruvate (CH_3_COCO_2_H), ATP, NADH and H_2_O [61,93]. Since glycolysis comprises two phases—one that consumes two molecules of ATP used for phosphorylation of one molecule of glucose, and a posterior one that produces four molecules of ATP—the net gain of the process is two molecules each of ATP, pyruvate, NADH and H_2_O. Pyruvate is a central intermediary in fermentation and its fate is determined by reducing equivalent disposal (electron transference in redox reactions). For example, pyruvate is decarboxylated to Acetyl-CoA by a hydrogenase, with the consequent production of hydrogen and CO_2_, which in turns reduces ferredoxin or CO_2_ to produce formate [93,94,95]. Oxidation of NADH to NAD transforms pyruvate into lactate in a reversible fashion or gives rise to succinate from reduction of oxaloacetate. Succinate is finally metabolized to propionate and lactate can be metabolized to acetate, butyrate or propionate [21,74,96]. Hemicelluloses also contain pentose, a sugar with five carbon atoms (C_5_H_10_O_5_), through which the pentose cycle gives rise to acetate and ribose 5-phosphate [74,97]. The latter, through anabolic reactions, leads to the production of acetyl-CoA, propionyl-CoA, pyruvate and oxaloacetate. In general, reoxidation of reduced cofactors carried out by hydrogenases also transfers electrons to H^+^ to form H_2_ (molecular hydrogen), which is intercepted by methanogens for the production of methane from CO_2_ and H_2_O [55,74,76,98]. Since propionate production, but not acetate or butyrate production, implies utilization of H (constitutes a H sink), the profile of VFAs is intimately linked to methane production [5,74,99]. Moreover, hydrogenotrophic acetogenic bacteria (homo-acetogens) use the Wood-Ljungdahl pathway to metabolize CO_2_ and H_2_ into acetate [75,100]. In Figure 1B we present a simplified description of the chemical reactions taking place in carbohydrate fermentation.

The aqueous medium in the rumen contains secreted bacterial enzymes and soluble nutrients like sugars, peptides and amino acids, which can be utilized by other bacteria that lack the ability to extract such nutrients [89]. However, microbes in the ruminal aqueous media (planktonic ones) likely do not play a pivotal role in fermentation and represent instead transient states mainly constituted by daughter cells that eventually will attach to a biofilm formed on feed particles [21,101]. A biofilm is a consortia of structurally and functionally organized bacteria formed by the activity of cellulolytic bacteria that produce mono- and disaccharide sugars, which are subsequently used to produce SCFA. One such SCFA (formate) is utilized by some Archaea, i.e., Methanobrevibacter species, for methanogenesis [101]. Methanogens utilize the hydrogen released during reverse oxidation of reduced cofactors and produce methane, thus contributing to reducing acidification of the rumen [55,74,76,96]. The syntrophic relationship established in biofilms favors fermentation efficiency and sustainability [90]. Evolution of the fermentative rumen environment attempts to maximize the production of energy for the host and its microbiome and has, obviously, not taken into consideration the anthropocentric inconvenience of methane emissions. Therefore, extensive livestock production systems represent a predicament wherein production of meat and dairy products, which is directly proportional to carbohydrates fermentation, is expected to be maximized, while emissions of greenhouse gasses like methane should be urgently reduced [54,102,103,104].

## 3. Role of *Prevotella* in Lignocellulose Processing

The rumen is a seemingly infinite collection of microbes that establish imbricate interactions both with their host and among themselves. As a whole, they produce SCFAs that are absorbed by the rumen epithelium for growth and the eventual production of meat and milk [99,105,106]. Despite their high level of organization, the microbiome configurations are unstable and influenced by host diet and by interspecies competition [1,23,40,54,105].

Energetic efficiency of animals refers to the percentage of ingested calories that are converted into animal products, e.g., meat and milk [107,108,109]. Genome-wide studies have shown that specific genomic regions are associated with energetic efficiency [110] but the predicted heritability is only moderate [111]. The remaining variability in feed efficiency among ruminants could be explained by the rumen microbiome [106,112,113]. Interestingly, Shabat and collaborators [106] found an inverse correlation between feed efficiency and microbiome richness. Highly efficient animals harbored higher concentrations of the SCFAs propionate, butyrate, valerate, and isovalerate as well as higher propionate-to-acetate ratio. Lower-efficiency animals were enriched in bacterial genes that represent a more diverse array of metabolic functions and also in methanogens archaea. The authors concluded that highly efficient animals process a less diverse array of metabolites and specialize in the production of fatty acids with a concomitant reduction in methane production [106]. Further results published by the same group revealed that proteomics data are more informative to delineate the actual microbiome metabolic profile than metagenomics data, thus fueling a long-standing debate concerning the intrinsic divergence between gene content and actual protein expression in metagenomics studies [114]. Nevertheless, these and a myriad of additional studies led to the proposal of a host genetics-microbiome axis as the master control mechanism of metabolism in ruminants [103,115].

Saccharification refers to the process whereby carbohydrates are hydrolyzed to component sugar by bacterial enzymes. In Bacteroidetes, like *Prevotella,* such enzymes are organized as PULs (Figure 1C), which are sets of co-regulated genes active in saccharification of complex carbohydrates (glycans) [65]. However, PULs are not exclusive to ruminants, but rather are widespread across species [3,64,65,66]. The central part of PULs corresponds to a starch utilization system (*Sus*; Figure 1C), best characterized in *Bacteroides thetaiotaomicron* but ubiquitous in Bacteroidetes [116]. A comprehensive collection of carbohydrate-active enzymes (CAZymes) is maintained at the PULDB site (www.cazy.org; accessed on 12 September 2022) [117] and the dbCAN-PUL site (https://bcb.unl.edu/dbCAN_PUL; accessed on 12 September 2022) [118]. Computational tools for prediction of PULs have also been developed [117,119,120]. At the time of writing, five classes of CAZymes are indexed in the CAZy database. Glycoside hydrolases (GH; EC 3.2.1.-; 173 families) hydrolyze glycosidic bonds between carbohydrates and other moieties. Glycosyl transferases (GT; EC 2.4.x.y; 115 families) catalyze glycosidic bonds between activated donor and specific acceptor molecules. Polysaccharide lyases (PL; EC 4.2.2.-; 42 families) cleave uronic acid-containing polysaccharides to produce unsaturated hexeuronic acid and a new reducing end. Carbohydrate esterases (CE; 20 families) catalyze the acylation of substituted saccharides. Lastly, there are 17 families of redox enzymes that perform functions auxiliary to CAZymes, e.g., lignin degradation [121].

*Prevotella* genomes contain many PULs that encode CAZymes specialized in the degradation of non-cellulosic plant fibers. So far, for *Prevotella* alone, the CAZy database contains entries for 9858 CAZymes which represent 191 CAZymes families in 85 bacterial strains. Of those, 22 correspond to carbohydrate binding enzymes; 12 are carbohydrate esterases; 116 are glycoside hydrolases; 26 are glycosyl transferases; and 15 are polysaccharide lyases. The five strains with the larger number of characterized CAZymes are *Prevotella* sp. E13-3 (267 CAZymes), *P. ruminicola* 23 (274 CAZymes), *P.* sp. E15-14 (278 CAZymes), *P. ruminicola* KHP1 (279 CAZymes) and *P. bryantii* B14 (310 CAZymes) [117,121]. However, we acknowledge that the number of CAZymes characterized in each strain may be a reflection of the anthropocentric interest they generate more than their actual CAZymes repertoire encoded in their genomes. A genetic toolbox for studying PULs in *Prevotella* has been developed [86] and it should shed light into the specific role of more *Prevotella* CAZymes in the years to come. PULs stratified a series of *Prevotella* species into specialists and generalists and exhibited considerable diversity within the generalists [3]. Accetto and Avguštin [122] characterized the PULs from 39 species and 50 genomes of *Prevotella* from diverse habitats. Approximately 1200 full-length SusD proteins, arranged in tandem with SusC proteins, were detected but the average number per species was quite variable, with species harboring a reduced number of SusD proteins exhibiting a proportional number of CAZymes. Out of 25 SusD protein groups, only three groups targeted hemicellulose or pectin, suggesting that *Prevotella* catalytic activity centers mainly on glycans and plant storage polysaccharides. On the basis of their SusD enzymes, the *Prevotella* species could be clustered into five groups, and the ruminal *Prevotella* clustered together with oral and gut species. Finally, six SusD protein groups included peptidases, evidencing a role in protein degradation and peptide transport [122]. An independent study reported similar results for *Prevotella bryantii*, which exhibited high catalytic affinity for plant storage and the cell wall polysaccharide β-glucan [123]. In enrichment cultures inoculated with ruminal content from cattle fed a total mixed ration, *Prevotella* poorly fermented hemicellulose, but turned dominant in cultures supplemented with xylan [14]. PULs containing Sus-like homologs have been reported in metagenomic studies in cattle [124,125,126,127], reindeer [126,128,129], moose [27] and sheep [126]. A Bacteroidetes phylotype (SRM-1) found in the reindeer metagenome was demonstrated to have glycoside hydrolase activity, apparently from members of the GH5 family of endoglucanases, and was shown to be active on β-glucans, xylans, xyloglucans and mannan substrates [128]. In general, it has been proposed that the efficiency of *Prevotella* for the digestion of xylans and pectins resides in a multitude of enzymes encoded in its gene clusters, especially the glycoside hydrolases xylanase GH10 and beta xylosidase GH43 [130]. One way of quantifying the relative contribution of specific bacterial taxa to carbohydrate fermentation is through comparisons of relative abundances of taxonomically preclassified sequences encoding CAZymes. In a deep sequenced metagenomics study in goats, *Prevotella* was the most abundant genus and was found to contribute 30% hemicellulases, 36% enzymes for lignocellulose pretreatment, 98.8% ferulolyl esterases and 71.1% acetylxylan esterases [11] of all sequences encoding such enzymes.

The ability of *Prevotella* to ferment carbohydrates is also reflected in the productivity of cattle. In a small cohort of lactating Holstein cows, it was reported that animals with high content protein and fat (HPF) in their milk harbored higher content of SCFAs (acetate, butyrate and propionate) in their ruminal fluid. Species in the genus *Prevotella* occupied 68.8% of all species identified and were more abundant in the HPF group. Among taxa found at higher abundance in the HPF group, functional annotation of bacterial genes using KEGG, eggNOG and CAZy databases, demonstrated an enrichment in functions pertaining to metabolism of carbohydrates, amino acids, pyruvate, insulin and lipids as well as transport [68]. Analogously, Conte and collaborators also reported a positive correlation between the abundance of *Prevotella* and lipid metabolism in cows [131]. In fattening yak in Tibet, *Prevotella* was the most abundant genus in the ruminal microbiome, and its abundance was positively and negatively correlated with polyunsaturated and saturated fatty acids, respectively [132].

In vitro, *Prevotella bryantii* has been reported to be involved in the de novo synthesis of amino acids from ammonia in the presence of amino acids and peptides, and the extent to which it is used for protein synthesis should have implications for carbohydrate fermentation efficiency [133]. The proteolytic ability of *Prevotella ruminicola* has been reported [70]. Also, in vitro experiments demonstrated that vitamin B12 increased propionate synthesis by *Prevotella ruminicola* 23, but in the absence of this vitamin the main synthesis product was succinate [67]. Interestingly, a positive correlation between the abundance of *Prevotella* and vitamin B12 was found in the rumen of Holstein cattle [134]. Together, the capability of *Prevotella* to synthesize lipids and amino acids, and its proteolytic capability confer great adaptability to the ruminal environment.

## 4. *Prevotella* and Its Association with Reduced Methane Emissions

Tropospheric greenhouse gasses, mainly methane, ozone and carbon dioxide regulate temperature and allow life on Earth to flourish, but such amenable temperatures have been fine-tuned over long periods of time. Research has shown that even small fluctuations in temperature, as a result of increasing greenhouse gas emissions, may have dramatic effects on life on Earth, e.g., on coral reefs, forests, biodiversity, etc. [135,136,137]. Following the industrial revolution, temperature has been on the rise, and such a trend has been exacerbated after the 1980s, mainly due to the use of fossil fuels and the expansion of agriculture, including livestock production systems [135]. Ruminants generate massive amounts of methane as a byproduct of carbohydrate fermentation [60,74,76,98,138]. Unlike gasses emitted during the combustion of fossil fuels, carbon emitted by ruminants is part of the carbon cycle since it was initially sequestered as CO_2_ by photosynthetic organisms. Specifically, the problem lies in the fact that ruminants belch out CH_4_ as one of the end products of ruminal microbial metabolism, which is a more potent greenhouse gas. CH_4_ is eventually converted back into CO_2_ but only after a period of ~20 years. It is generally accepted that ruminants contribute approximately 20% of global anthropogenic methane [139,140]. Methane oxidation in the troposphere contributes to the formation of ozone and carbon dioxide and therefore boosts global warming [72,141]. Although methanogenesis metabolizes only a handful of substrates, it is a very sophisticated mechanism that includes three different pathways and more than 200 genes [142]. In order to reduce the environmental footprint of livestock systems, it is imperative to implement production practices that lower methane emissions. The rumen microbiome stands out as one of the most promising targets for such interventions. Moreover, ruminal methane production also represents a loss of energy [102,143]; therefore, microbiome configurations that reduce methane production have the potential to both reduce the environmental footprint and increase energetic efficiency of production animals.

Although part of ruminal methane is generated via methylotrophic methanogenesis [144,145], the majority arises from hydrogenotrophic methanogenesis using H_2_ and CO_2_ as substrates [23,146]. In the latter pathway, the archaea genus *Methanobrevibacter* appears dominant, and its abundance is favored by a rich forage diet because it promotes synthesis of ATP [147]. Interestingly, the abundance of methanogenic archaea in the rumen holds only a weak correlation with methane emissions [36,102,103]. On the other hand, a series of studies in ruminants using a halogenated methane analog, bromochloromethane (BCM), which inhibits methanogenesis, showed that increases in ruminal H_2_ and H_2_ emissions were accompanied by an increase in the synthesis of propionate and isovalerate and by expanded populations of *Prevotella* spp. and *Fibrobacter succinogenes* [148,149,150,151]. Prevotella could sequester H by the succinate or the acrylate pathways that ferment sugars or lactate, respectively [67,152]. Although it is unlikely that BCM administration will be used as a practice to mitigate methane emissions—because it is itself considered a greenhouse gas—those experiments revealed that some microorganisms could efficiently divert the H_2_ pool away from methanogenesis.

A series of metagenomics studies have outlined a strong correlation between the abundance of *Prevotella* and lower methane emissions. For instance, in a cohort of Colombian buffalo, it was reported that animals with lower emissions of methane harbored higher densities of *P. ruminicola*, *P. bryantii, P. brevis* and 22 other *Prevotella* species. The authors hypothesized that *Prevotella* could contribute to reductions in methane emissions by diverting H towards the production of propionic acid and away from methanogenesis. Interestingly, no differences in the abundance of methanogenic archaea were observed between animals with low and high methane emissions. Metabolic profiling revealed that among the top *Prevotella* enzymes that increased in abundance were methylmalonyl CoA mutase (MCM), NADP-specific glutamate dehydrogenase (NADP-GDH), a sulfite exporter TauE/SafE family protein, and a FeS cluster assembly ATPase SufC protein [36]. Because data in this study capture in great detail what several studies suggest, we summarize in Figure 2 the prevalence of many *Prevotella* species (green bars in left panel) and *Prevotella* enzymes (green bars in right panel) in animals exhibiting lower methane emissions. Interestingly, vitamin B12 is a cofactor of MCM [153], and increased abundance of *Prevotella* was associated with augmented concentrations of vitamin B12 [134], which in turn influenced propionate synthesis in *Prevotella ruminicola* 23 [67]. In feedlot lambs, liver vitamin B12 concentration correlated with higher content of propionate metabolizing enzymes, but exogenous application of vitamin B2 did not accentuate such correlation [154]. Thus, supplementation of vitamin B12 is unlikely a good strategy to promote propionate biosynthesis as a means to lower methane production, although more investigations are still required to rule out such a possibility. The majority of studies concerning feed processing efficiency and its relationship with methane production have been conducted on dairy cows, buffalo or goats. In a study involving small cohorts of Holstein or Jersey steers under the same diet regime, it was found that methane production and VFA concentrations (mainly acetate and butyrate) were significantly higher in the Jersey cohort, while ruminal pH was lower. A series of *Prevotella* species, including *P. micans*, *P. copri*, *P. oris*, *P. baroniae* were found at a significantly higher density in the Holstein cohort, which produced less methane. Only *P. shahii* was found at reduced abundance [155].

A variety of studies have reported that the cashew nut shell liquid (CNSL), a phenolic subproduct of cashew nut production, has the potential to reduce ruminal methane emissions through the modification of the microbiome [156,157,158,159,160]. In a small cohort of Holstein cows, animals were subjected to a diet containing concentrate and hay (60:40) for four weeks (control time point) and then to the same diet supplemented with pellets containing 4 g/100 kg of body weight per day of CNLS (CNLS time point). In one of the trials, methane production per unit of dry matter was reduced by 38.3% and energy loss decreased from 9.7% to 6.1%. Bacterial species related to the production of formate (*Ruminococcus flavefaciens*, *Butyrivibrio fibrisolvens*, and *Treponema bryantii*) decreased in abundance while others related to propionate production (*Prevotella ruminicola*, *Selenomonas ruminantium*, *Anaerovibrio lipolytica*, and *Succinivibrio dextrinosolvens*) increased in abundance [160]. These experiments need validation in larger cohorts but highlight an important point: not only do propionate-producing species like *Prevotella ruminicola* contribute to a reduction in methane production, but the abundance of such bacteria is boosted by the suppression of methane generation. In other words, a feedback loop that progressively increases the relative abundance of propionate-producing bacteria and at the same time reduces methane emissions could be established by using chemical suppressors of methanogenesis and parallel strategies to foster abundance of such taxa. Conversely, it was reported that supplementation of nitrate led to a reduction of methane emissions in lactating cows and an increase in acetate to propionate ratio with a concomitant decrease in abundance of *Prevotella* [161].

In a factorial experimental design, buffaloes and bovines were fed two different roughage concentrate dietary mixes (70:30, low concentrate, LC; 40:60, high concentrate, HC) and their microbial diversity, ruminal methane emissions and nutrient utilization were determined. HC led to reduced methane emissions. Bacteroidetes increased their abundance in the HC diet, and *Prevotella* was the dominant genus in the rumen with higher relative abundance in bovines (52%) than in buffalo (32%); however the HC diet also increased *Prevotella* abundance in buffalo [162]. Yet, reductions in methane production are not exclusively associated with *Prevotella*. In studies on Holstein dairy cows whose diet was supplemented with two methane-mitigating agents (grape marc or a mixture of lipids and tannins) a series of contigs resembling *Faecalibacterium* sequences were found to be differentially accumulated in animals that received methane-mitigating agents. In an independent cohort, it was found that the abundance of such contigs correlated with methane production. No differential abundance of *Prevotella* was reported in that study [163].

Another major player in ruminal methane production is host genetics. Studies on ruminal content transplantation where modifications of the host microbiome structure and milk production efficiency of animals were only transient [164,165], suggested a role of the host genotype on microbiome composition. Host genotype has also been reported to influence CH_4_ emissions [166,167]. Thus, interest has arisen in the relative contribution of the microbiome and the host genotype on methanogenesis and the relationships among them. Difford and collaborators [168] assessed the hypothesis that the microbiome composition is heritable and that methane emissions are influenced by both the genome of the host and its microbiome. In a cohort of 750 lactating Holstein dairy cows from farms in Denmark, methane emissions were quite variable. There was a 41% mean difference between the top and bottom 10% emitters (519.28 ± 28.5 g/d vs. 303.8 ± 11.9 g/d, respectively). The vast majority of OTU’s found in >50% of cows corresponded to the phylum Bacteroidetes (72.2%). Using a linear mixed model, it was found that the archaeal genus *Methanobrevibacter* had a statistically significant heritability (h^2^) (0.22 ± 0.09). Host genetics and ruminal microbes explained 21% and 13% of the variation observed in methane production, respectively [168]. Similar results were obtained using Bayesian modeling, with even a smaller proportion of the variance in methane emissions explained by the microbiome [169]. The take-home message of these studies is that both host genetics and microbiome composition should be taken into consideration for designing strategies to mitigate ruminal methane emissions. Inside the ruminal microbiome, *Prevotella* appears a strong candidate to integrate methane mitigation strategies due to its frequent association with lower methane emissions and its high relative abundance in the rumen.

## 5. Concluding Remarks

*Prevotella* appears as a major player in ruminal metabolism. Several attributes of such a bacterium vindicate its preeminence. Namely, it is remarkably abundant in many species of ruminants; one of its fermentation products is propionic acid and has, therefore, the potential to compete with methanogenesis and archaea for hydrogen utilization. Consequently, its abundance has been found inversely correlated with methane emissions. Given the urgency to reduce greenhouse gas emissions, *Prevotella* has the potential to be used as an antimethanogenic agent. Several questions remain unanswered in that respect. For example, what is the appropriate formulation to supplement livestock feedstuff with *Prevotella*? Would exogenously administered *Prevotella* persist in the ruminal microbiome of receptor animals? Experiments involving administration of probiotic formulations that include *Prevotella* or transplantation of ruminal content from animals harboring high density of *Prevotella* will shed light into those questions.

## Figures and Tables

**Figure 1 microorganisms-11-00001-f001:**
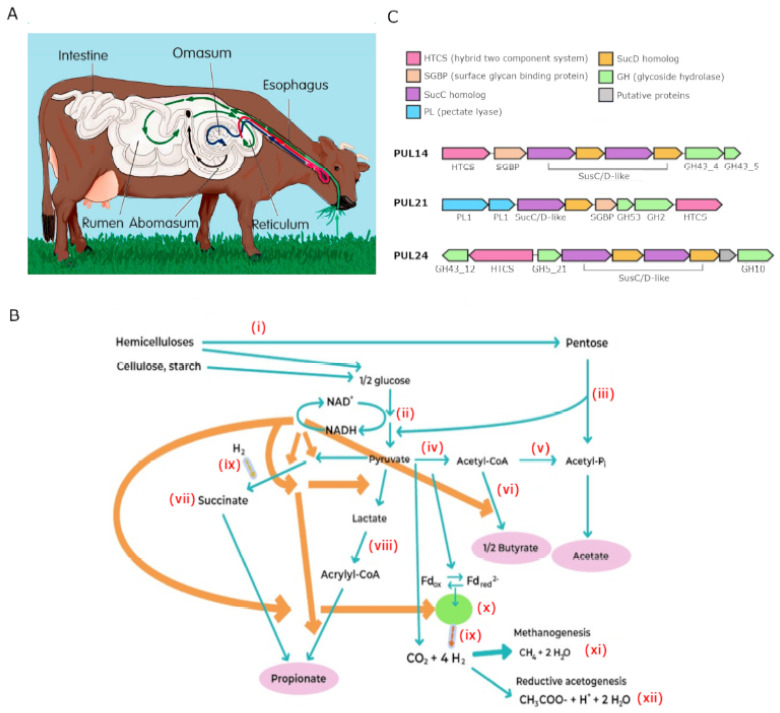
Ruminal carbohydrates fermentation. (**A**) Ruminants lack enzymes for digestion of cellulose contained in the cell wall of plant material. Therefore, digestion of carbohydrates starts with initial mastication and passage of feedstuff to the rumen (green arrow), where cellulolytic bacteria partially degrade plant cell walls. Semi-digested material flows then to the reticulum, which facilitates regurgitation (red arrow) and rechewing of feed particles. Remasticated, finer feed particles are mobilized to the omasum (blue arrow). Finally, feedstuff passes to the abomasum (black arrow), which is considered the true stomach where an acidic pH facilitates digestion of microbial and plant proteins. (**B**) Simplified schematics of carbohydrate fermentation. (**i**) Breakdown of polysaccharides to monosaccharides; (**ii**) A glucose molecule is oxidized into two molecules of pyruvate with concomitant reduction of NAD^+^ to NADH; (**iii**) Pentose metabolism through the pentose cycle and transketolase cleavage; (**iv**) Pyruvate oxidative decarboxylation with production of CO_2_, reduced ferredoxin and acetyl-CoA. An alternative reaction releases formate instead CO_2_ and reduced ferredoxin; (**v**) Acetate production; (**vi**) Butyrate production; (**vii**) Propionate production via the succinate pathway; (**viii**) Propionate production via acrylate pathway. (**ix**) Interspecies hydrogen transference; (**x**) Production of molecular hydrogen through electron confurcation; (**xi**) Hydrogenotrophic methanogenesis; (**xii**) Reductive acetogenesis; which is a smaller H_2_ sink than (**xi**). Not all reactants or products are shown. Figure 1B was modified from [85]. Orange thick arrows indicate points where the redox couple (NAD+ and NADH) promote chemical reactions. (**C**) Genetic architectures of some PUL loci in *P. copri*, the central SusC/D proteins, the hybrid two component system (HTCS), glycoside hydrolases (GH), pectate lyases (PL) and surface glycan binding proteins (SGBP) are depicted. Figure 1C was modified from [86].

**Figure 2 microorganisms-11-00001-f002:**
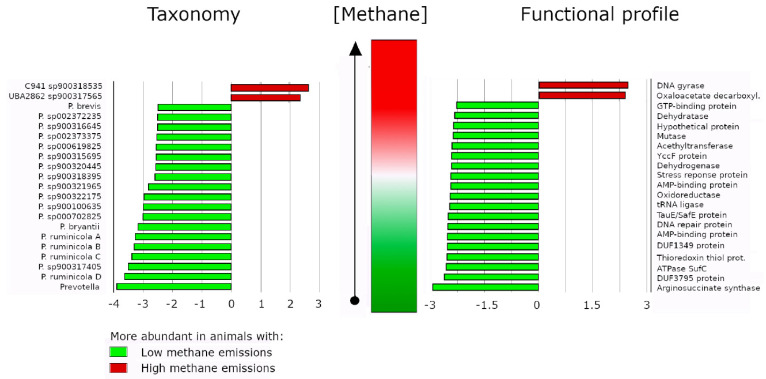
Inverse correlation between *Prevotella* abundance and methane emissions. Data from Aguilar-Marin et al., 2020 [36]. Results from linear discriminant analysis with software LEfSe. All presented results had a *p*-value < 0.05 and showed that many species of *Prevotella* (left panel) or proteins from *Prevotella* (right panel) were more abundant in animals that exhibited lower methane emissions (green bars). Red bars depict taxa or proteins that were more abundant in animals with high methane emissions. P: *Prevotella*. Suffixes A, B, C and D in *Prevotella ruminicola* species are arbitrary and are used here only to denote that four different strains of such a species were found differentially accumulated. For the sake of space, names of proteins in the right panel were arbitrarily abbreviated. All green bars correspond to *Prevotella* proteins while the two red bars correspond to proteins from *Bacteroides ovatus* (DNA gyrase, top bar) and [*Ruminoccocacea bacterium* AB4001] (Oxoloacetate decarboxylase). Full names of proteins can be found in Figure 6 of Aguilar-Marin et al., 2022 [36].

## Data Availability

Not applicable.

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
