# Peer review of "Prevotella: A Key Player in Ruminal Metabolism"

_microorganisms, 2022, doi:10.3390/microorganisms11010001_

Round 1
Reviewer 1 Report
Well written and interesting review on the role of Prevotella in the rumen. It would have been interesting a little final paragraphs dealing with the different possibilities (speculatives) to influence Prevotella population in rumen microbiome in order to reduce methan emissions.
Only some minor comments:
Introduction line 3 "Hallella" in italic
Ruminal carbohydrate fermentation Line "induces" instead of *induce"
Page 4 line 2 "ingests" instead of "ingest"
Page 7 line "the majority arises from " instead of "the majority arises by"
Author Response
It would have been interesting a little final paragraphs dealing with the different possibilities (speculatives) to influence Prevotella population in rumen microbiome in order to reduce methan emissions.
We have now included a new section called Concluding remarks putting forward the suggestion of Reviewer #1.
Only some minor comments:
Introduction line 3 "Hallella" in italic
Done
Ruminal carbohydrate fermentation Line "induces" instead of *induce"
Done
Page 4 line 2 "ingests" instead of "ingest"
Done
Page 7 line "the majority arises from " instead of "the majority arises by"
Done
Reviewer 2 Report
Reviewer’s comments on the manuscript by Betancur-Murillo et al. entitled: Prevotella: a key player in ruminal metabolism.
Manuscript ID: microorganisms-2028990
November, 2022.
Prevotella is a very versatile microbe capable of processing a wide range of proteins and polysaccharides. This manuscript review summarizes the biochemistry of carbohydrate fermentation and the essential role of Prevotella in lignocellulose processing and its association with reduced methane emissions. The authors investigated an interesting topic, good written. In my opinion, this manuscript can acceptance. Specific Comments are as follows:
-How about conclusion in this manuscript, could add conclusion at the end of the manuscript?
- I suggest draw flow chart (picture) for Prevotella and its association with reduced methane emissions.
Author Response
-How about conclusion in this manuscript, could add conclusion at the end of the manuscript?
Thank you very much for the suggestion. We including a final section now called Concluding remarks that certainly improves the manuscript.
- I suggest draw flow chart (picture) for Prevotella and its association with reduced methane emissions
We added a second figure where we summarize some data from Aguilar-Marin et al., 2022 that reflect the inverse relationship between Prevotella abundance and methane emissions. Thanks for this suggestion.